# Association of pneumococcal carriage in infants with the risk of carriage among their contacts in Nha Trang, Vietnam: A nested cross-sectional survey

**George Qian**[1]☯*, **Michiko Toizumi**[2]☯, **Sam Clifford**[1], **Lien Thuy Le**[3], **Tasos Papastylianou**[4], **Catherine Satzke**[5], **Billy Quilty**[1], **Chihiro Iwasaki**[2], **Noriko Kitamura**[2], **Mizuki Takegata**[2], **Minh Xuan Bui**[6], **Hien Anh Thi Nguyen**[7], **Duc Anh Dang**[7], **Albert Jan van Hoek**[8], **Lay Myint Yoshida**[2‡], **Stefan Flasche**[1]*‡

**1** Centre for Mathematical Modelling of Infectious Diseases, London School of Hygiene & Tropical Medicine, London, United Kingdom, **2** Institute of Tropical Medicine, Nagasaki University, Nagasaki, Japan, **3** Department of Bacteriology, the Pasteur Institute in Nha Trang, Nha Trang, Vietnam, **4** School of Computer Science and Electronic Engineering, University of Essex, Colchester, United Kingdom, **5** Translational Microbiology Group, Murdoch Children's Research Institute at the Royal Children's Hospital, University of Melbourne, Parkville, Australia, **6** Khanh Hoa Health Service, Nha Trang, Vietnam, **7** National Institute of Hygiene and Epidemiology, Hanoi, Vietnam, **8** Centre for Infectious Disease Control, National Institute for Public Health and the Environment, Bilthoven, the Netherlands

☯ These authors contributed equally to this work.
‡ LMY and SF also contributed equally to this work.
* George.Qian@lshtm.ac.uk (GQ); Stefan.Flasche@lshtm.ac.uk (SF)

**Data Availability Statement:** All relevant data can be found in the following repository: https://github.

## Abstract

### Background

Infants are at highest risk of pneumococcal disease. Their added protection through herd effects is a key part in the considerations on optimal pneumococcal vaccination strategies. Yet, little is currently known about the main transmission pathways to this vulnerable age group. Hence, this study investigates pneumococcal transmission routes to infants in the coastal city of Nha Trang, Vietnam.

### Methods and findings

In October 2018, we conducted a nested cross-sectional contact and pneumococcal carriage survey in randomly selected 4- to 11-month-old infants across all 27 communes of Nha Trang. Bayesian logistic regression models were used to estimate age specific carriage prevalence in the population, a proxy for the probability that a contact of a given age could lead to pneumococcal exposure for the infant. We used another Bayesian logistic regression model to estimate the correlation between infant carriage and the probability that at least one of their reported contacts carried pneumococci, controlling for age and locality. In total, 1,583 infants between 4 and 13 months old participated, with 7,428 contacts reported. Few infants (5%, or 86 infants) attended day care, and carriage prevalence was 22% (353 infants). Most infants (61%, or 966 infants) had less than a 25% probability to have had close contact with a pneumococcal carrier on the surveyed day. Pneumococcal infection

com/GeorgeYQian/Nha-Trang-Contact-Study-Data-and-Code.

**Funding:** SF, SC and the infant contact survey were supported by a Sir Henry Dale Fellowship awarded to SF jointly funded by the Wellcome Trust and the Royal Society (Grant number: 208812/Z/17/Z). LMY, MT, CS, GQ, BQ and the pneumococcal carriage survey of the Nha Trang PCV reduced dosing study is funded by the Bill & Melinda Gates Foundation (Grant number: OPP1139859). The Nha Trang population-based cohort study was supported by AMED under Grant Number JP21wm0125006. CS was supported by an Australian NHMRC Career Development Fellowship and a Veski Inspiring Women Fellowship. MCRI was supported by the Victorian Government's Operational Infrastructure Support Program. The funders had no role in study design, data collection and analysis, decision to publish, or preparation of the manuscript.

**Competing interests:** CS is Lead investigator on a Merck Investigator Studies Program grant funded by MSD on pneumococcal serotype epidemiology in children with empyema. CS is Investigator on a clinical research collaboration on PCV vaccination in Mongolia.

**Abbreviations:** CI, confidence interval; Ct, cycle threshold; PCV, pneumococcal conjugate vaccine; PEI, potential exposure index; qPCR, quantitative PCR; SARS-CoV-2, Severe Acute Respiratory Syndrome Coronavirus 2; STROBE, Strengthening the Reporting of Observational Studies in Epidemiology; VIF, variance inflation factor.

risk and contact behaviour were highly correlated: If adjusted for age and locality, the odds of an infant's carriage increased by 22% (95% confidence interval (CI): 15 to 29) per 10 percentage points increase in the probability to have had close contact with at least 1 pneumococcal carrier. Moreover, 2- to 6-year-old children contributed 51% (95% CI: 39 to 63) to the total direct pneumococcal exposure risks to infants in this setting. The main limitation of this study is that exposure risk was assessed indirectly by the age-dependent propensity for carriage of a contact and not by assessing carriage of such contacts directly.

## Conclusions

In this study, we observed that cross-sectional contact and infection studies could help identify pneumococcal transmission routes and that preschool-age children may be the largest reservoir for pneumococcal transmission to infants in Nha Trang, Vietnam.

## Author summary

### Why was this study done?

- There is little information in the current literature on pneumococcal transmission to infants.

- Since individuals in this age group are at major risk of pneumococcal disease and at potentially reduced protection in reduced dose pneumococcal vaccination schedules, we carried out a study to identify their close contacts and likely routes of transmission.

### What did the researchers do and find?

- We identified all 7,428 physical contacts of 1,583 infants in Nha Trang, a coastal city in Vietnam.

- We found that there was a high correlation between infant infection and their probability of contact with at least one pneumococcal carrier on the surveyed day.

- Preschool-age children contributed around half of the total direct pneumococcal exposure risk to infants in Nha Trang. Around 80% of the exposure risk came from contact with household members.

### What do these findings mean?

- Social contact surveys can help to identify possible transmission pathways.

- Infants in Nha Trang are likely to receive substantial indirect protection against pneumococcal infection from vaccinated older siblings.

## Introduction

Mathematical modelling is a key part of the evidence synthesis process that informs public health policy for infectious diseases and their mitigation, not least evident in its role in the Severe Acute Respiratory Syndrome Coronavirus 2 (SARS-CoV-2) pandemic [1,2]. An essential part of such infectious disease models is their adequate reflection of transmission in the community studied. For respiratory pathogens, social contacts and their age structure have been used predominantly as the key source of heterogeneity in infectious disease spread, and, hence, the accurate measurement of social contacts as a proxy for potential transmission events has been fundamental for the validity of such models [3,4].

While social contacts are a seemingly obvious proxy for transmission risk for pathogens transmitted via droplets or aerosols from the respiratory tract, there is limited direct evidence that the common definition of a transmission relevant contact, i.e., a 2-way conversation or skin-to-skin contact, is indeed a major risk factor for infection. Nested contact and infection studies in Fiji and Uganda have shown that the frequency of physical (skin-to-skin) household contacts of longer duration (>1 hour) was associated with higher rates of pneumococcal carriage [5,6], and a prospective cohort study in the United Kingdom showed that adults hospitalised with community acquired pneumonia had increased odds for close child contact within the 4 weeks prior to their illness [7]. Furthermore, during the SARS-CoV-2 pandemic, changes in contact behaviour as measured by social contact surveys have enabled real-time estimation of the impact of non-pharmaceutical interventions on pandemic transmission intensity [8,9].

Infants are typically at high risk for severe respiratory disease and are targeted for either direct or indirect vaccine protection. Their exposure to disease through their social contacts can also be an important guide for selecting different dosing schedules for these vaccines, including pneumococcal conjugate vaccines (PCVs) [10]. However, there is little evidence on what constitutes a transmission relevant contact for infants or the frequency and distribution of such contacts. Most contact studies include few infants [3,9] or are conducted in specific environments that capture only a portion of their contacts [11]. The most detailed study to date, which studied the contact patterns of 115 infants in the UK, found that almost two-thirds of infant contacts were not with household members, highlighting the potential importance of non-household transmission routes for infant infection [12].

We report the results of a large infant contact study nested into a cross-sectional pneumococcal carriage survey in Nha Trang, Vietnam. We investigate the correlation of infant exposure risk, approximated by reported social contacts and the prevalence of infection in such, with their risk of pneumococcal infection and the spatial and demographic structure of infant contacts and exposure in this setting.

## Methods

### Study population

The study was conducted in Nha Trang, a coastal city in south-central Vietnam with a total population of just over 426,958 and under-five population of 28,495 in the 2018 census (personal communication with Dr. Minh Xuan Bui, Khanh Hoa Health Service, October 2, 2021—see S3 File). Nha Trang city consists of 27 communes of similar population size. Each commune has a commune health centre, providing a range of basic health services. Similarly, educational services including nurseries and schools are largely provided on a commune basis.

In October 2016, a cluster randomised controlled trial was initiated to evaluate alternative PCV dosing schedules [13]. In the 24 communes that were allocated to receive PCV, routine infant vaccination was initiated according to the allocated schedule: 2 priming doses and 1

booster dose (2p+1), 3p+0, 1p+1, or 0p+1, and a catch-up campaign including children under 3 years old was conducted. The primary endpoint of the trial is dependent on pneumococcal carriage, and, as such, this is monitored in each commune through annual cross-sectional nasopharyngeal carriage surveys in 60 infants (4 months to 11 months), 60 toddlers (14 months to 23 months), randomly selected from an administrative population list, and their respective main carers. Completion rate of routine PCV vaccination in the community was 81.2%, 88.5%, 97.4%, and 87.0%, in the 2p+1, 1p+1, 0p+1, and 3p+0 arms, respectively, using the 18 months population as denominator for the 2p+1, 1p+1, and 0p+1 arms and the 12 months population for the 3p+0 arm, in 2018 when the infant contact survey was conducted. Due to serotype replacement, the trial did not affect overall carriage rates in the Nha Trang population.

## Study design

In October 2018, 2 years after the start of the trial, an infant contact survey was nested into the cross-sectional carriage study. All infants enrolled for the carriage study were also eligible for enrolment into the contact survey.

For each commune health centre, a staff member was identified and trained to conduct the contact survey in their local commune. During an initial home visit, written informed consent was obtained for both the infant contact and carriage studies from parents or guardians before any collection of data or specimen.

Subsequently, a background questionnaire was filled in together with the parent or guardian on behalf of the infant, collecting information including sex, age, household composition, and the infant's mobility (see S1 File for the full questionnaire). The parent or guardian was asked to monitor their infant's contacts the following day, and another home visit was scheduled for the day after. For that visit, the interviewer would aid the carer's memory by discussing their day and noting initials of the infant's contacts. Subsequently, information on those contacts' age, sex, duration, and location was filled in (see S2 File for the full questionnaire). A contact was defined as skin-to-skin contact with the infant, because the common conversational contact definition would be inappropriate to apply to infants and because physical contacts, as a proxy for close contacts, have been shown to better correlate with pneumococcal carriage risk [5,6].

Consent to participate in the contact study and the nasopharyngeal carriage study was sought from the carers by home visit. The date of keeping a contact diary was assigned, and field workers conducted a second visit 1 or 2 days after to record the contact information based on the diary. Participants were also given an appointment with the commune health centre during which a nasopharyngeal swab from the infant was obtained by a study nurse. Sample collection, handling and storage were performed according to the World Health Organization guidelines [14]. At the Pasteur Institute of Nha Trang, DNA extractions were performed on a QIAcube HT instrument using the QIAmp 96 DNA QIAcube HT Kit (QIAGEN, Hilden, Germany) with a lysis buffer (20 mg/ml lysozyme, 20 mM Tris/HCl, 2 mM EDTA, 1% v/v Triton) and RNase A treatment, as previously described [15]. Real-time quantitative PCR (qPCR) to detect the autolysin-encoding gene (*lytA*) of *Streptococcus pneumoniae* was performed. Final reaction volumes of 25 μl were run on an Applied Biosystems 7500 Fast real-time PCR instrument with 5 μl of template DNA, TaqMan GeneExpression Mastermix (Applied Biosystems, Massachusetts, USA), and 200 nM of primers and probe (forward primer: 5′-ACGCAATCTAGCAGATGAAGCA-3′, reverse primer: 5′-TCGTGCGTT TTAATTCCAGCT-3′, and *lytA* probe: 5′ FAM -TGCCGAAAACGCTTGATACAGGGAG- 3′ BHQ1) [16]. *lytA* positive (cycle threshold (Ct) value < 35) or equivocal (Ct value 35 to 40) samples were cultured on sheep blood agar plates with 5 μg/ml of gentamicin. Samples with alpha-hemolytic growth were harvested and DNA was extracted from the bacterial pellet on a

QIAcube HT instrument as described previously [17]. Pneumococcal serotyping was performed by DNA microarray at the Murdoch Children's Research Institute using Senti-SPv1.5 microarrays (BUGS Bioscience, London, UK) [18].

Pneumococcal carriage was defined as samples that were both *lytA* positive/equivocal and culture positive and subsequently confirmed by microarray. A small number of samples that were *lytA* qPCR positive but were nonculturable (*n* = 5) or had technical issues (*n* = 4) were also deemed pneumococcal positive.

This study is reported as per the Strengthening the Reporting of Observational Studies in Epidemiology (STROBE) guideline (S1 STROBE Checklist).

## Statistical analyses

The infant contact study is a cross-sectional study, and no prospective analysis plan exists. The plan for the current study combining contact and carriage data was conceived during a collaborative meeting that happened after the infant contact and pneumococcal carriage surveys were conducted, but without being informed by the data. We used negative binomial regression to estimate crude rate ratios of the mean number of contacts of participants for each characteristic of the participants, as defined in Table 1. We also calculated rate ratios adjusted for age (categorical: 4 to 7 months and 8 to 11 months of age) and for clustering of participants into communes.

After completion of the nested contact survey, we also examined the association of infant infection and their contact behaviour but, in an extension to previous studies, also accounted for the risk of infection in each contact. We used logistic regression to estimate the crude ratio of odds for a contact being made outside the commune of residence, for each characteristic of the participants. In an adjusted analysis, we included the age group (4 to 7 and 8 to 11 months of age: the first 4 months and the second 4 months of the target age range) and day care attendance as covariates and accounted for clustering of participants into communes.

To explore whether an infant's contact patterns are correlated with their probability to carry pneumococcus, we created the potential exposure index (PEI). For each infant participating in the study, PEI estimates the probability that at least one of the reported contacts was carrying pneumococcus and hence that the infant was exposed to the pneumococcus on that day at least once. We define PEI as

$$PEI = 1 - \Pi_i(1 - p_i), \tag{1}$$

where $p_i$ represents the probability that the *i*th contact of the respective study participant is a pneumococcal carrier. We assume that $p_i$ is solely dependent on the contact's age. To estimate age-dependent probabilities for pneumococcal carriage in the study setting, we performed Bayesian penalised B-spline regression on logit-transformed carriage probability, obtained from previously reported low-granularity age-stratified pneumococcal carriage across age groups from Nha Trang [19]. These data were collected in October 2006 from 519 individuals in a cross-sectional survey. The age group stratifications were yearly from ages until 5 years old and then 6 to 10 years old, 11 to 17 years old, 18 to 30 years old, 31 to 49 years old, and over 50 years old. We visually validated our estimates against carriage prevalence observed in young children in the trial. The PEI estimate is based on the contact patterns recorded for a single day but were interpreted as a proxy for the general contact behaviour of the respective infant. The Mann-Whitney U test was used to assess whether there was a significant difference in average PEI value in carriers and noncarriers.

We used a hierarchical Bayesian logistic regression model (see Table B in S4 File) to estimate the correlation of PEI (covariate) and pneumococcal infection (outcome). Based on selection via a Bayesian belief network (see Fig A in S4 File), we included age in months as a

**Table 1. Mean number of contacts reported for infants and effect of each infant's characteristic on incidence of contacts, estimated using a negative binomial regression model.**

| Characteristic | | | Number | Mean (SD) | Incidence rate ratio | p-Value | Adjusted incidence rate ratio* | p-Value |
|---|---|---|---|---|---|---|---|---|
| | Total | | 1,583 | 4.7 (1.8) | | | | |
| Demographics | | | | | | | | |
| | Sex | | | | | | | |
| | | Male | 871 | 4.7 (1.8) | reference | 0.376 | reference | |
| | | Female | 712 | 4.6 (1.8) | 0.98 (0.94 to 1.03) | | 0.98 (0.94 to 1.01) | 0.233 |
| | Age at the time of enrolment (months) | | | | | | | |
| | | 4 to 7 months | 616 | 4.7 (1.8) | reference | 0.56 | | |
| | | 8 to 11 months | 967 | 4.7 (1.8) | 1.01 (0.97 to 1.06) | | | |
| Family | | | | | | | | |
| | Siblings in the household | | | | | | | |
| | | No sibling | 666 | 4.6 (1.9) | reference | 0.256 | reference | |
| | | One or more siblings | 917 | 4.7 (1.7) | 1.03 (0.98 to 1.08) | | 1.03 (0.99 to 1.06) | 0.103 |
| | Number of people in the household | | | | | | | |
| | | 2 to 4 | 611 | 4.1 (1.7) | reference | <0.001 | reference | |
| | | >4 | 972 | 5.1 (1.7) | 1.25 (1.19 to 1.31) | | 1.25 (1.19 to 1.30) | <0.001 |
| | Caretaker currently in a paid employment | | | | | | | |
| | | No | 1,056 | 4.7 (1.9) | reference | 0.673 | reference | |
| | | Yes | 527 | 4.7 (1.6) | 1.01 (0.96 to 1.06) | | 1.01 (0.96 to 1.06) | 0.725 |
| | Highest level of education in the household | | | | | | | |
| | | Secondary school or lower | 268 | 4.8 (1.8) | reference | 0.568 | reference | |
| | | Degree | 1,315 | 4.7 (1.8) | 0.98 (0.93 to 1.04) | | 0.98 (0.92 to 1.05) | 0.582 |
| Infant's activity | | | | | | | | |
| | Sit | | | | | | | |
| | | Yes | 1,334 | 4.7 (1.8) | 1.04 (0.98 to 1.11) | 0.177 | 1.05 (0.99 to 1.11) | 0.094 |
| | | No | 249 | 4.5 (1.7) | reference | | reference | |
| | Crawl | | | | | | | |
| | | Yes | 873 | 4.7 (1.8) | 1.01 (0.96 to 1.05) | 0.752 | 1.00 (0.95 to 1.05) | 0.977 |
| | | No | 710 | 4.7 (1.8) | reference | | reference | |
| | Walk | | | | | | | |
| | | Yes | 186 | 4.7 (2.0) | 0.99 (0.93 to 1.07) | 0.863 | 1.10 (0.98 to 1.24) | 0.647 |
| | | No | 1,397 | 4.7 (1.8) | reference | | reference | |
| Mobility | | | | | | | | |
| | Transport available for immediate travel | | | | | | | |
| | Bicycle | | | | | | | |
| | | Yes | 31 | 4.5 (1.4) | 0.95 (0.80 to 1.12) | 0.532 | 0.95 (0.86 to 1.04) | 0.251 |
| | | No | 1,552 | 4.7 (1.8) | reference | | reference | |
| | Motorbike | | | | | | | |
| | | Yes | 1,549 | 4.7 (1.8) | 1.15 (0.97 to 1.36) | 0.1 | 1.15 (1.06 to 1.25) | 0.001 |
| | | No | 34 | 4.1 (1.2) | reference | | reference | |
| | Car | | | | | | | |
| | | Yes | 99 | 4.9 (2.1) | 1.04 (0.95 to 1.14) | 0.376 | 1.04 (0.97 to 1.13) | 0.286 |
| | | No | 1,484 | 4.7 (1.8) | reference | | reference | |
| | Walk | | | | | | | |
| | | Yes | 108 | 5.1 (2.0) | 1.1 (1.01 to 1.20) | 0.027 | 1.10 (0.98 to 1.24) | 0.105 |
| | | No | 1,475 | 4.7 (1.8) | reference | | reference | |
| | Public transportation | | | | | | | |

(*Continued*)

**Table 1.** (Continued)

| Characteristic | | | Number | Mean (SD) | Incidence rate ratio | p-Value | Adjusted incidence rate ratio* | p-Value |
|---|---|---|---|---|---|---|---|---|
| | | Yes | 40 | 4.8 (1.8) | 1.03 (0.89 to 1.19) | 0.695 | 1.03 (0.90 to 1.18) | 0.65 |
| | | No | 1,543 | 4.7 (1.8) | reference | | reference | |
| | Number of times caretaker left the commune in the last 7 days | | | | | | | |
| | | 0 to 2 | 872 | 4.5 (1.7) | reference | 0.002 | reference | 0.001 |
| | | 3 or more | 711 | 4.9 (1.9) | 1.07 (1.03 to 1.12) | | 1.07 (1.03 to 1.12) | |
| | Number of times infant left the commune in the last 7 days | | | | | | | |
| | | 0 | 788 | 4.6 (1.7) | reference | 0.012 | reference | 0.004 |
| | | 1 or more | 795 | 4.8 (1.8) | 1.06 (1.01 to 1.11) | | 1.06 (1.02 to 1.10) | |
| | Day care attendance | | | | | | | |
| | | Yes | 86 | 5.0 (1.5) | 1.07 (0.97 to 1.18) | 0.16 | 1.07 (1.00 to 1.14) | 0.053 |
| | | No | 1,497 | 4.7 (1.8) | reference | | reference | |
| Pneumococcus carriage | | | | | | | | |
| | | Yes | 353 | 4.8 (1.8) | 1.03 (0.97 to 1.09) | 0.299 | 1.03 (0.98 to 1.08) | 0.284 |
| | | No | 1,229 | 4.7 (1.8) | | | | |

*Incidence rate ratios adjusted for age group, considering clustering in each commune.

covariate and the resident commune as a random effect. Hence, our logistic regression model was

$$logit(p_i) = b_0 + C_i + b_1 E_i + b_2 A_i, \qquad (2)$$

where

- $p_i$ is the probability that the $i$th infant in the study is a pneumococcal carrier;

- $b_0$ is the model intercept under a single-level regression model (fixed effects only);

- $C_i$ is the random effect on the intercept due to the influence of the commune to which the $i$th individual belongs;

- $b_1$ is the coefficient corresponding to the effect of PEI;

- $E_i$ is the value of PEI, for the $i$th individual;

- $b_2$ is the coefficient corresponding to the effect of the infant's age; and

- $A_i$ is the age of the $i$th individual in months.

The model was implemented in JAGS [20]. The uncertainty in the individual level estimates for PEI in this analysis was retained from the posterior estimates in the previous regression (namely, the cubic spline on age) by assuming a normal distribution for each year of age and drawing samples from this distribution when calculating the posterior samples of PEI. We chose to assume a normal distribution for simplicity and since, by visual inspection, the distribution of carriage rates across all ages appeared to follow a Gaussian (see Fig C in S4 File). Subsequently, this individual-level uncertainty in PEI is further propagated by assuming a beta distribution for these PEI estimates, since PEI is bounded between 0 and 1, and each individual's PEI distribution is right skewed (see Fig B in S4 File); this distribution then serves as an input to the logistic regression predicting carriage status. Potential correlations among the

predictor variables were checked before model fitting using the variance inflation factor (VIF) [21].

The total pneumococcal exposure to an infant on the day that contacts were reported was approximated by the sum of each reported contact's age specific probability of pneumococcal carriage. The contribution of a specific age group to that total was calculated as the pneumococcal exposure from contacts of that age group divided by the total exposure. This can be generalised to calculate the contribution of specific age groups to the total pneumococcal exposure across all infants by including all contacts in the study.

The descriptive statistics were done in STATA version 14.0 [22]. The analyses on the association of contact patterns and pneumococcal carriage were conducted in R 4.0.2 [23] and RJags [24]. The code is available on https://github.com/GeorgeYQian/Nha-Trang-Contact-Study-Data-and-Code.

### Ethical approval

Ethical approval for the study was granted by the respective ethics boards of the National Institute of Hygiene and Epidemiology in Hanoi, Vietnam (4405/QD-BYT) and the London School of Hygiene & Tropical Medicine (15881).

## Results

### Characteristics of study participants and their contacts

The contact study was conducted between October 19 and November 7, 2018. Overall, 1,583 infants aged between 4 and 13 months (median: 9 and interquartile range: 7 to 11 months) were enrolled in both the contact survey and the carriage survey (Table 1). All children were aged 11 months or younger at the time of enrolment, although 122 children were 12 or 13 months by the time of data collection. An additional 11 infants were enrolled in the carriage survey but did not consent to the contact survey because it was not convenient for the carer; these infants were not included in any of the following analyses (response rate was 99%). There were no missing data in the contact survey except that *S. pneumoniae* carriage status could not be determined in one infant due to technical reasons. This infant was excluded from the analysis of pneumococcal carriage. Over half (917, 58%) of infants had siblings living in the same household and the average household size was 4.9 (SD 1.0). The majority of infants were able to sit on their own (84%), many could crawl (55%), and some could walk (12%). Most families (98%) had a motorbike; other means of transport were rarely used. Moreover, 50% of infants did not leave their commune of residence during the preceding week. Few infants (5%) attended day care. Pneumococcal carriage prevalence was 22% and increased from 17% at 4 to 7 months to 26% at 8 to 11 months.

A total of 7,428 contacts aged between 0 and 100 years (median: 33 years) were reported (Fig 1). Contacts mostly occurred at home (89%) and with family members (76%). They were mostly with other children, parents, and grandparents and lasted for more than 1 hour (55%). Few contacts (5%) were reported to occur outside the commune of residence or with other infants (0.4%) (Table 2).

### Factors affecting the number of contacts

The number of skin-to-skin contacts an infant made on the day before the survey ranged from 1 to 15, and the mean was 4.7 (SD 1.8). Infants who lived in households with more than 4 people were reported to have 25% (95% confidence interval (CI): 19% to 30%) more contacts than infants in smaller households (adjusted incidence rate ratio 1.25, 95% CI: 1.19 to 1.30, *p*-value

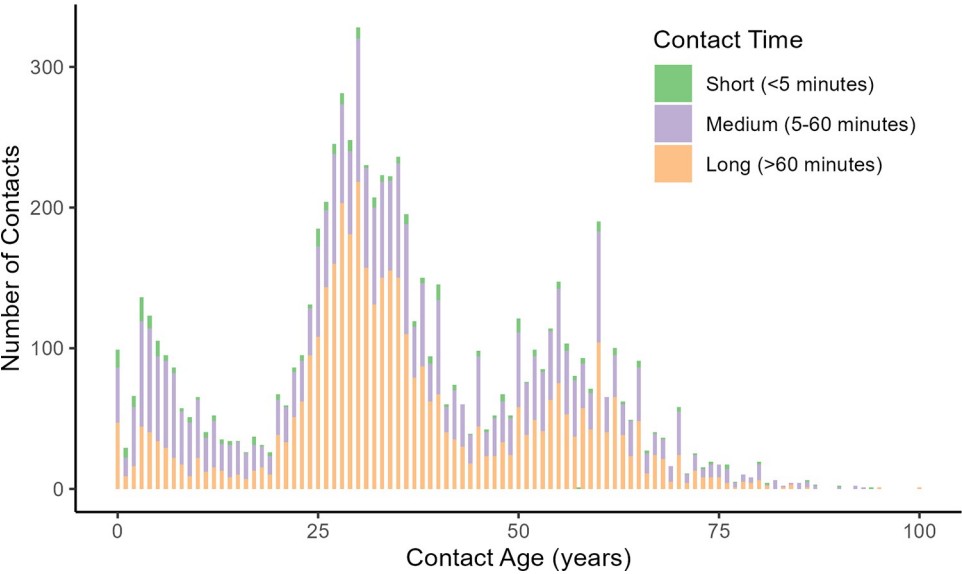

**Fig 1. The number of infant contacts.** Reported skin-to-skin contacts are shown disaggregated into annual age groups and stratified by contact duration.

<0.001). The infant's ability to crawl or sit was not found to be associated with their number of contacts; however, infants who had travelled beyond commune borders in the week preceding the survey were reported to have 6% (95% CI: 2% to 10%) more contacts (adjusted incidence rate ratio 1.06, 95% CI: 1.02 to 1.10, *p*-value 0.004). Infants who were found to carry pneumococci were reported to have slightly more contacts; however, this was not statistically significant (adjusted incidence rate ratio 1.03, 95% CI: 0.98 to 1.08, *p*-value 0.284) (Table 1).

## Factors affecting the location of contacts

About 10% of infants had at least one contact outside of their commune of residence (Table A in S4 File). Infants at least 8 months old and those with caretakers in current employment were 67% (95% CI: 16 to 140) and 67% (95% CI: 19 to 134) more likely (according to the crude odds ratios 1.67, 95% CI: 1.16 to 2.40, *p*-value 0.006 and 1.67, 95% CI: 1.19 to 2.34, *p*-value 0.003) to have a contact outside their commune, respectively. Also, pneumococcal carriers were more likely to have had contact outside the commune; however, this was no longer the case if adjusted for day care centre attendance (adjusted odds ratio 1.25, 95% CI: 0.92 to 1.71, *p*-value 0.159), which was strongly associated with contacts outside the commune (adjusted odds ratio 6.56, 95% CI: 3.92 to 10.98, *p*-value <0.001). Infant-to-infant contact was rare; only 2% of infants were reported to have contacted another infant in another commune. Of such infant-to-infant contacts, the majority (19/31, 61%) happened in day care.

## Validating contacts as a risk factor for carriage

Modelled carriage prevalence in Nha Trang based on carriage observations from a previous study [19] increased rapidly in infancy and peaked at about 4 years of age. Secondary school–age children and particularly adults were rarely carrying pneumococcus; about 10% and less than 5% respectively were infected. Carriage rates observed in 2008 and in 2018 were similar in the age groups observed in both surveys (Fig 2).

We estimate that most (61%) infants had less than a 25% probability of having had close contact with a pneumococcal carrier on the surveyed day; however, for 22% of infants, the

**Table 2. Characteristics of infants' contacts.**

| Characteristics | | Total, *n* = 7,428 (100%) |
|---|---|---|
| Sex | | |
| | Male | 2,988 (40.2) |
| | Female | 4,353 (58.6) |
| | Group contact | 87 (1.2) |
| Family member | | |
| | Yes | 5,672 (76.4) |
| | No | 1,756 (23.6) |
| Place of contact (first contact) | | |
| | At home | 6,600 (88.9) |
| | Day care | 134 (1.8) |
| | Transport | 1 (0.0) |
| | Office | 17 (0.2) |
| | Leisure | 115 (1.6) |
| | Other | 561 (7.6) |
| Contact occurred in the infant's residence commune | | |
| | Yes | 7,074 (95.2) |
| | No | 354 (4.8) |
| Contact duration | | |
| | <5 minutes | 331 (4.5) |
| | 6 minutes to 1 hour | 3,009 (40.5) |
| | >1 hour | 4,088 (55.0) |
| Contact frequency | | |
| | Daily or almost daily | 6,466 (87.1) |
| | Once or twice a week | 715 (9.6) |
| | Once or twice a month | 162 (2.2) |
| | Less than once a month | 52 (0.7) |
| | Never met before | 33 (0.4) |

probability of exposure was much higher at 50% to 70%. The estimated PEI for each infant was associated with substantial uncertainty (Fig B in S4 File). The median PEI value for pneumococcal carriers was 27% (95% CI: 5.1% to 70%), compared with 15% (95% CI: 4.9% to 61%) for noncarriers, and, from the Mann–Whitney U test, there was a statistically significant difference between PEI values from the 2 groups ($p < 0.001$).

When adjusting for age and locality, we find that the probability of exposure to pneumococci on the surveyed day and the risk of pneumococcal carriage were highly correlated: for every 10 percentage points increase in PEI, the odds of pneumococcal carriage increased by 22% (95% CI: 15 to 29), i.e., the odds of pneumococcal carriage were 7.1 (95% CI: 4.1 to 12.3) times higher in infants who had at least one effective contact with a pneumococcal carrier (PEI = 1), compared to those who had not (Table C in S4 File).

## Age-specific infant exposure to pneumococci

Children between 1 and 4 years of age contributed 38% (95% CI: 22% to 56%) of the total pneumococcal exposure to infants; children 5 to 9 and adults 21 to 40 year olds contributed 22% (95% CI: 16 to 29) and 20% (95% CI: 9.2 to 32), respectively. We estimate that 2- to 6-year-old children are key pneumococcal transmitters to infants, contributing 51% (95% CI: 39 to 63) of the total exposure and that infection risk from other infants or older children and

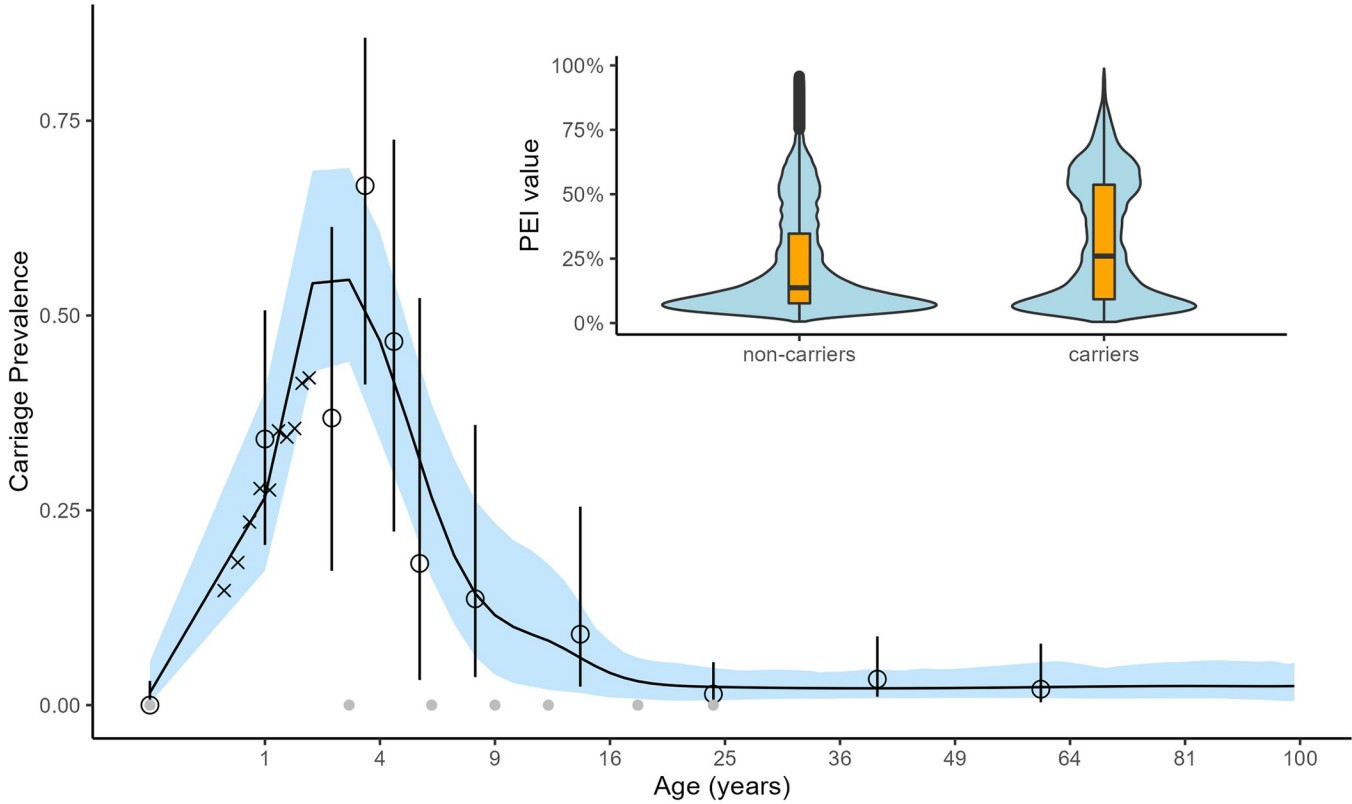

**Fig 2. Age-stratified carriage prevalence in Nha Trang.** Grey dots indicate the positions of knots for the spline and the open circles and their corresponding 95% binomial CIs the carriage prevalence data that the model was fitted to. The black line and the region highlighted in blue represents the model's median and 95% quantile estimates. The "x" symbols are added for visual comparison and indicate the carriage prevalence observed in the infant survey. To aid visualisation, we used quadratic scaling of the x-axis. Inset: the distribution of estimated PEI values in infants without (left) and with (right) pneumococcal carriage. CI, confidence interval; PEI, potential exposure index.

adults is likely low in this setting (Fig 3). The majority of potential pneumococcal exposures (80%) originated from household contacts.

## Discussion

We described in detail the social contacts of infants in Nha Trang, Vietnam. Unlike other age groups, infant contacts are not age assortative and happen largely within the household. We show that close social contacts, as measured in contact surveys on a specific day, and in combination with age specific infection probabilities of such contacts, are highly correlated with the risk of pneumococcal infection of infants in Nha Trang. Thus, we estimate that 1 to 4 year olds and 5 to 9 year olds contribute most (39% (95% CI: 32% to 47%) and 22% (95% CI: 16 to 29), respectively) to the infection pressure to infants in this setting and that 80% of infections are acquired from household contacts.

We have previously shown that pneumococcal infection risk is correlated with the frequency of physical contacts [5,6]. Here, we expand that notion by including the probability that contacts are infected with pneumococci, similar to how age-stratified dynamic mathematical models are constructed [3,25,26], and thus provide direct evidence for the value of social contact structure as a proxy for disease transmission routes in mathematical models. Furthermore, we suggest that the combination of social contacts and age-dependent infection probabilities provides a simple and useful method to identify likely transmission routes, without the

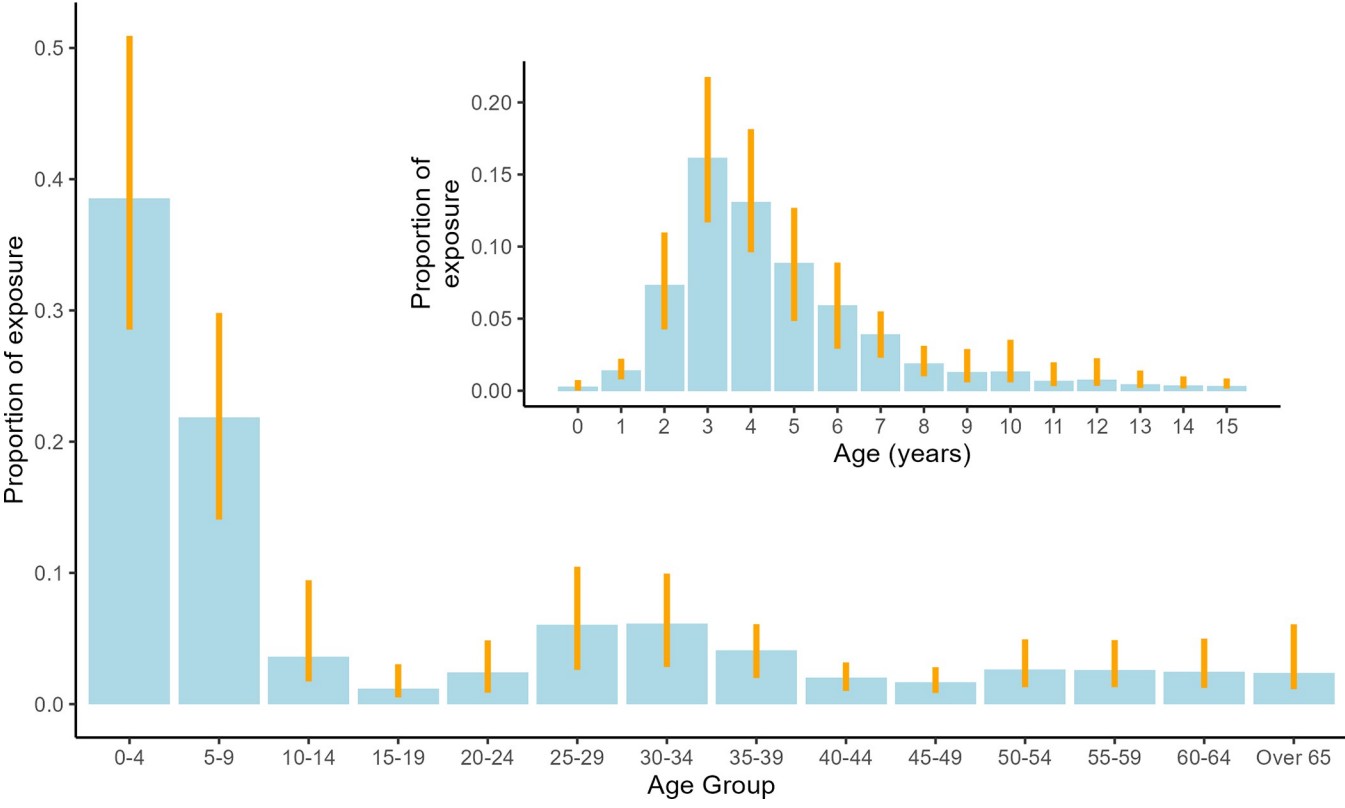

**Fig 3. The contribution of different age groups towards the total exposure of infants to pneumococcus.** Bars show the relative contribution of an age group to the total pneumococcal exposure (number of close contacts and their propensity to carry pneumococci) of infants in Nha Trang. Error bars indicate 95% credible intervals. Inset: the contribution of up to 15 year olds to total pneumococcal exposure of infants in Nha Trang.

need for more complex mathematical modelling or the need for longitudinal data collection [27,28].

Similar to their key role in pneumococcal transmission generally [25,29–31], we find that 2- to 6-year-old children are the likely main direct transmitters of pneumococcal infection to infants. This has important implications for the optimal design of pneumococcal vaccination strategies, particularly those that aim to sustain herd protection while reducing the number of doses given in infancy [10]. While the exact duration of protection from PCVs remains relatively uncertain [32], booster dose schedules may induce longer lasting protection and hence may be preferred in settings with high carriage prevalence in older children and hence a potentially high contribution to the pneumococcal infection pressure from that age group [33].

We find that 76% of contacts and 80% of potential pneumococcal exposures to the infants in this study were with members of their households. In contrast, in the UK, the household contacts had a less dominant role and 11-week- to 12-month-old infants were reported to have less than 50% of their contacts at home [12]. This illustrates the importance of context in the evaluation of transmission routes for infants. For example, while in the UK many children will attend nursery from 6 to 8 months of age, in Vietnam, publicly funded nurseries will only accept children older than 12 months. Thus, infant infection routes may be substantially different in settings where childcare responsibilities are shared with older generations or siblings.

Our study suggests that pneumococcal exposure risk to infants stems mostly from household contacts and particularly siblings. In combination with the commune organised schooling system in Vietnam this suggests that the commune is a major spatial entity in determining

pneumococcal infection in infants in Nha Trang. The main mode of transport for families in this community was travel by motorcycle, with the vast majority of families owning and using one for travel. The infants were mostly able to sit but fewer were able to crawl or walk (55% and 12%, respectively), and so for individuals who had not achieved these milestones, contact would likely be initiated by parents or siblings.

A limitation of this study is that there is large uncertainty around the carriage rates in teenagers. However, direct contact with teenagers was rare for our study population and hence the associated uncertainty in infection probability has little impact on the overall results. In settings where teenagers share childcare responsibilities and where high pneumococcal infection rates extend well into teenage years [34–36], such contacts may contribute a larger part to the infection pressure on infants. A further limitation is that contacts were only surveyed on a single day and that carriage status was only assessed about a week after that. The strong correlation between contact behaviour and pneumococcal carriage in our study suggests that, indeed, the contact behaviour on a given day is somewhat representative for the general contact behaviour of that infant. This is an important validation for the adequacy of social contact surveys for informing mathematical transmission models advising public health decision making. Our survey period did not include any major holidays such as the Tết Nguyên Đán, or Lunar New Year. Such events may well change contact patterns and accelerate the spread of pneumococci in infants. It is also worth noting that our estimates for the relative proportions of exposure to infants represent only the direct contributions of each age group. Our study does not allow us to ascertain additional indirect effects of exposure that would arise when considering second generation (or beyond) exposure patterns.

In summary, we provide direct evidence that for pneumococci, and thus likely other respiratory pathogens, social contact surveys indeed add a crucial part in our understanding of transmission pathways. We find that 2- to 6-year-old children are the most likely source of pneumococcal infection in infants in Nha Trang. Thus, it adds support to catch-up vaccination of preschoolers to accelerate impact upon introduction as well as supports schedules aimed at extending direct protection into this age group to establish pronounced herd protection. Similar studies in other settings can help evaluate local pneumococcal transmission routes and hence provide crucial evidence for the discussion on optimal dosing schedules for PCVs.

## Supporting information

**S1 STROBE Checklist. STROBE checklist of the infant pneumococcal carriage and contact studies.** STROBE, Strengthening the Reporting of Observational Studies in Epidemiology. (PDF)

**S1 File. Background questionnaire.** (PDF)

**S2 File. Contact questionnaire.** (PDF)

**S3 File. Personal communications permission form.** (PDF)

**S4 File. Supporting information to the paper.** (DOCX)

## Acknowledgments

We would like to thank the staff of the Khanh Hoa Health Service and the Pasteur Institute in Nha Trang for their support in conducting the contact survey and the pneumococcal carriage survey. We thank Kim Mulholland, Monica Nation, Jason Hinds, and the team from MCRI for supporting the microbiology for the study and particularly the microarray testing and data interpretation.

## Author Contributions

**Conceptualization:** Albert Jan van Hoek, Lay Myint Yoshida, Stefan Flasche.

**Data curation:** Catherine Satzke.

**Formal analysis:** George Qian, Michiko Toizumi, Sam Clifford.

**Investigation:** Michiko Toizumi.

**Software:** Tasos Papastylianou, Billy Quilty.

**Supervision:** Stefan Flasche.

**Writing – original draft:** George Qian, Michiko Toizumi, Stefan Flasche.

**Writing – review & editing:** George Qian, Michiko Toizumi, Sam Clifford, Lien Thuy Le, Tasos Papastylianou, Catherine Satzke, Billy Quilty, Chihiro Iwasaki, Noriko Kitamura, Mizuki Takegata, Minh Xuan Bui, Hien Anh Thi Nguyen, Duc Anh Dang, Albert Jan van Hoek, Lay Myint Yoshida, Stefan Flasche.

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
