## [Editor Report · Decision Letter 0]

6 Jul 2021

Dear Dr Qian, 

Thank you for submitting your manuscript entitled "Pneumococcal exposure routes for infants, a nested cross-sectional survey in Nha Trang, Vietnam" for consideration by PLOS Medicine.

Your manuscript has now been evaluated by the PLOS Medicine editorial staff and I am writing to let you know that we would like to send your submission out for external peer review.

Please re-submit your manuscript within two working days, i.e. by Jul 08 2021 11:59PM.

Kind regards,

Caitlin Moyer, Ph.D.

Associate Editor

PLOS Medicine

---

## [Decision Letter · Decision Letter 1]

15 Feb 2022

Dear Dr. Qian,

Thank you very much for submitting your manuscript "Pneumococcal exposure routes for infants, a nested cross-sectional survey in Nha Trang, Vietnam" (PMEDICINE-D-21-02876R1) for consideration at PLOS Medicine. 

Your paper was evaluated by a senior editor and discussed among all the editors here. It was also discussed with an academic editor with relevant expertise, and sent to three independent reviewers, including a statistical reviewer. The reviews are appended at the bottom of this email and any accompanying reviewer attachments can be seen via the link below:

[LINK]

In light of these reviews, I am afraid that we will not be able to accept the manuscript for publication in the journal in its current form, but we would like to consider a revised version that addresses the reviewers' and editors' comments. Obviously we cannot make any decision about publication until we have seen the revised manuscript and your response, and we plan to seek re-review by one or more of the reviewers. 

We expect to receive your revised manuscript by Mar 08 2022 11:59PM. Please email us (plosmedicine@plos.org) if you have any questions or concerns.

We look forward to receiving your revised manuscript. 

Sincerely,

Caitlin Moyer, Ph.D.

Associate Editor

PLOS Medicine

plosmedicine.org

1. Title: Please revise your title according to PLOS Medicine's style. Your title must be nondeclarative and not a question. It should begin with main concept if possible. "Effect of" should be used only if causality can be inferred, i.e., for an RCT. Please place the study design ("A randomized controlled trial," "A retrospective study," "A modelling study," etc.) in the subtitle (ie, after a colon).

2. Throughout: Please include line numbers with the revised version.

3. Abstract: Methods and Findings: Please explicitly mention the years during which the study took place and main outcome measures. Please specify the ages of the infants.

4. Abstract: Methods and Findings: In the last sentence of the Abstract Methods and Findings section, please describe the main limitation(s) of the study's methodology.

5. Abstract: Conclusions: Please address the study implications without overreaching what can be concluded from the data; beginning with the phrase "In this study, we observed ..." may be useful.

6. Author Summary: At this stage, we ask that you include a short, non-technical Author Summary of your research to make findings accessible to a wide audience that includes both scientists and non-scientists. The Author Summary should immediately follow the Abstract in your revised manuscript. This text is subject to editorial change and should be distinct from the scientific abstract. Please see our author guidelines for more information: https://journals.plos.org/plosmedicine/s/revising-your-manuscript#loc-author-summary

7. Methods: “The study was conducted in Nha Trang, a coastal city in south-central Vietnam with a total population of just over 426,958 and under-five population of 28,495 in the 2018 census (personal communication with Khanh Hoa Health Service).” 

The information may be cited in the text as a personal communication with the author if the author provides written permission to be named. Please provide the name of the individual, the affiliation, and date of communication. The individual named must provide written permission to be named. Alternatively, please provide a different appropriate reference.

8. Methods: “In October 2016 a cluster randomised controlled trial was initiated to evaluate alternative pneumococcal conjugate vaccine (PCV) dosing schedules [13].” It would be helpful to describe relevant details of the trial, including recruitment and enrollment of participants. The Methods indicate that 24 of 27 communes were allocated to receive PCV vaccination. Please comment on how this would be expected to impact infant carriage and the PEI estimates, for example.

9. Methods: “In October 2018, two years after the start of the trial, an infant contact survey was nested into the cross-sectional carriage study.” Please provide some information on the PCV vaccine schedule and coverage at the time when the survey was carried out. Please define how the infant population was sampled for the carriage study/contact survey. Please report the response rates, clarify the dates when the contact survey was conducted.

10. Methods: “In an adjusted analysis we included the age group (4-7 and 8-11 months of age)” Please explain the rationale for the categorization of ages into these groups.

11. Methods: “Microbiological culture was subsequently attempted for lytA positive (cycle threshold (Ct) value < 35) or equivocal (Ct value 35-40) samples. Pneumococcal carriage was defined as samples that were both lytA positive/equivocal and culture positive, and subsequently confirmed by microarray [15] .” Please briefly describe the methods for qPCR detection, including primers, as well as for microbiological culture and microarray confirmation.

12. Methods: “We used negative binomial regression to estimate crude rate ratios of the mean number of contacts of participants in different population subgroups.” Please define the population subgroups.

13. Methods: “To estimate age dependent probabilities for pneumococcal carriage in the study setting, we performed Bayesian Penalised B-Spline Regression on logit-transformed carriage probability, obtained from previously reported low-granularity age-stratified pneumococcal carriage across age groups from Nha Trang [16]. We compared our estimates against carriage prevalence observed in young children in the trial. The PEI estimate is based on the contact patterns recorded for a single day but were interpreted as a proxy for the general contact behaviour of the respective infant.” 

Please provide additional description of the low-granularity age-stratified carriage data from reference 16 (including years when those data were obtained), as well as more description of how age dependent probabilities obtained were compared to the carriage prevalence observed in young children in the trial (including a reference for those data).

14. Methods: Prospective analysis plan: Did your study have a prospective protocol or analysis plan? Please state this (either way) early in the Methods section.

15. Methods: STROBE: Please report your study according to the relevant guideline, which can be found here: http://www.equator-network.org/. Please ensure that the study is reported according to the STROBE guideline, if that is most appropriate, and include the completed STROBE checklist as Supporting Information. Please add the following statement, or similar, to the Methods: "This study is reported as per the Strengthening the Reporting of Observational Studies in Epidemiology (STROBE) guideline (S1 Checklist)."

16. Results: Please present numerators and denominators for percentages, at least in the Tables (not necessarily each time they're mentioned).

17. Results: For the results described, please report in the text the results of all statistical analyses, with both CIs and p values.

18. Results: “Modelled carriage prevalence in Nha Trang based on carriage observations from a previous study…” please provide a reference for the study.

19. Discussion: Please present and organize the Discussion as follows: a short, clear summary of the article's findings; what the study adds to existing research and where and why the results may differ from previous research; strengths and limitations of the study; implications and next steps for research, clinical practice, and/or public policy; one-paragraph conclusion.

20. Tables and Supporting Information Tables: Please report p values in addition to confidence intervals presented for all analyses. Please also present the unadjusted results for all analyses. In the legends, please note statistical tests used, and factors adjusted for in the analyses.

21. Table S4: Please define “DIC” and “PEI” in the legend.

22. Table S5: Please define “PEI” and explain “Commune ID” in the legend.

23. Table S6: Please define “ PEI” and “VIF” in the legend.

24. Figure S4: Please define “PEI” in the legend.

25. Figure S5: Please define “MCMC” and “PEI” in the legend.

Comments from the reviewers:

Reviewer #1: In the proposed paper, the authors report the results of a large pneumococcus carriage and contact tracing study from Vietnam. They develop an individual-level 'potential exposure index' based on the age-structure of contacts. The descriptions of the number, location and type of contacts is an important addition to the literature, as is the age-specific probabilities of carriage. I only have a few comments.

The authors claim that on average pneumococcal carriers had a higher PEI than non-carriers - however, the uncertainty with these number is (very) high. I do not believe the authors can make this statement - instead they should do a statistical test to see if the PEI levels were different between the two groups.

It would be useful to understand the distributions of PEIs - overall and within each age group. At a population level each age group is assumed to have a normally distributed PEI values - but it is unclear how appropriate this is.

The authors state that individuals that almost certainly had an exposure (PE=1) had a high odds of being positive - however, we do not know how common that is and what approximately 1 means (>0.95?). I assume that this implies that carriage within an age group is randomly distributed and that e.g., in a classroom of children each will have the same probability of being positive and their risks will not be correlated between members of the class - this should be stated and discussed. 

I would move the logistic regression equation to the main text - it will make it clear what approach the authors have taken.

The authors assume a beta distribution for the PEI distributions - it wasn't clear how appropriate this assumption was - a histogram of the posterior with a beta distribution would help.

It would be good to include a discussion of ways to improve PEI estimation - in particular in reducing uncertainty. For example, would looking at serotype/GPSC distribution help?

Reviewer #2: In this study, the authors combine a high-quality dataset of pneumococcal carriage with detailed survey data to infer the contribution of different age groups to transmission of pneumococcus. The survey data are used to develop a novel index that quantifies an individual child's risk of exposure given their contacts and the prevalence of pneumococcus in those contacts (by age). This is a clean, intuitive approach that aids in understanding the importance of different age groups for transmission. The authors also provide a well-documented Github repository, which should allow others to replicate the analyses in other settings.

My only major comment is about the interpretation and directionality of the exposure index. For instance the authors "estimate that 2 to 6 year old children are key pneumococcal transmitters to infants, contributing 51% (95% CI: 39-63) of the total exposure, and that infection risk from other infants or older children and adults is likely low in this setting". Thinking about population-level effects, 51% of exposure is directly attributed to kids in that age group. But if the 2-6 year olds are also responsible for a large fraction of transmission to older children and adults, the effect would be even greater because the 2-6 year olds transmit to adults, who then transmit to infants. For example, in an extreme situation where all transmission in the population originated with 2-6 year olds, Is there a way to account for this type of dependency? If not, it should be clear that these estimates, which capture the direct contribution from the age group, represent a lower bound for the total contribution of that age group. 

The authors could also add some discussion about the implications of their findings for vaccination strategies. What is the potential policy response to knowing that 2-6 year olds are important for transmission? Catch-up campaigns? different dose timing?

Reviewer #3: This is an interesting paper combining pneumococcal carriage and social contact data to assess the risk of pneumococcal transmission as a function of social contacts among infants, building on previous studies in Uganda and Fiji. 

The authors have used an elegant statistical approach to estimate the potential exposure to a pneumococcal carrier (and hence infection risk) based on knowledge about the social contacts of infants and the probability of carriage of their contacts. The method allows to further explore the type of social contacts that matter for pneumococcal transmission. The methods are clear and the paper is well written

I have few very comments on the paper:

1. In their calculation of the Potential Exposure Index, authors are using age-specific carriage probabilities. Have they looked at sex too, particularly among adult populations (caregivers, mothers). While the carriage prevalence in that setting is very low across all age groups, and the carriage difference between adult age groups is likely small too, it would be useful to explore

2. It is a little unclear from the description whether there were variations among the study populations in vaccination schedule and/or coverage, given that this was nested within a trial, and whether carriage estimates differed between the different datasets used (pre-trial, trial). Please clarify

[LINK]

---

## [Decision Letter · Decision Letter 2]

26 Apr 2022

Dear Dr. Qian,

Thank you very much for re-submitting your manuscript "Pneumococcal exposure routes for infants, a nested cross-sectional survey in Nha Trang, Vietnam" (PMEDICINE-D-21-02876R2) for review by PLOS Medicine.

I have discussed the paper with my colleagues and the academic editor and it was also seen again by three reviewers. I am pleased to say that provided the remaining editorial and production issues are dealt with we are planning to accept the paper for publication in the journal.

[LINK]

We look forward to receiving the revised manuscript by Apr 28 2022 11:59PM.   

Sincerely,

Caitlin Moyer, Ph.D.

Associate Editor 

PLOS Medicine

plosmedicine.org

Requests from Editors:

1. Title: Please place the study design in the subtitle (ie, after a colon) and please update this in the manuscript submission system in addition to the main text. We suggest revising to: “Association of pneumococcal carriage in infants with the risk of carriage among their contacts in Nha Trang, Vietnam: A nested cross-sectional survey”

2. Abstract: Background: Please add a final sentence that clearly states the study question.

3. Abstract: Methods and Findings, Line 10: Please add the sentence punctuation: “In October 2018, we conducted a nested cross-sectional contact and pneumococcal carriage survey in randomly selected 4 to 11 months old infants across all 27 communes of Nha Trang, Vietnam.”

4. Abstract: Methods and Findings: Line 16: Please report the actual numbers, in addition to percentages.

5. Author summary: Please reformat the author summary, providing 2-3 single sentence bullet points under each of the following headings. Why Was This Study Done?; What Did the Researchers Do and Find?; What Do These Findings Mean? Please see https://journals.plos.org/plosmedicine/s/revising-your-manuscript#loc-author-summary for guidelines and examples.

We suggest reformatting as follows, or similar:

Why was this study done?

-There is little information in the current literature on pneumococcal transmission to infants. -Since individuals in this age group are at major risk of pneumococcal disease and at potentially reduced protection in reduced dose pneumococcal vaccination schedules, we carried out a study to identify their close contacts and likely routes of transmission.

What did the researchers do and find?

-We identified all 7428 physical contacts of 1583 infants in Nha Trang, a coastal city in Vietnam. -We found that there was a high correlation between infant infection and their probability of contact with at least one pneumococcal carrier on the surveyed day.

-Preschool-age children contributed around half of the total direct pneumococcal exposure risk to infants in Nha Trang. Around 80% of the exposure risk came from contact with household members.

What do these findings mean?

-Social contact surveys can help to identify possible transmission pathways.

- Infants in Nha Trang may be likely to receive substantial indirect protection against pneumococcal infection from vaccinated older siblings.

6. Methods: Line 93-94: Please also include a reference to the personal communication letter in the Supporting Information.

7. Methods: Line 123, 128, and throughout: The terms gender and sex are not interchangeable (as discussed in http://www.who.int/gender/whatisgender/en/ ); please use the appropriate term.

8. Methods: Line 134-135: Please specify if this is “written informed consent” for both contact and carriage studies.

9. Methods: Line 250: Please change this to “National Institute of Hygiene and Epidemiology in Hanoi, Vietnam”

10. Methods: Line 163: Prospective analysis plan: Please explicitly state that the study had no prospective analysis plan. Please explicitly state when these analyses were planned, and when and why any data-driven changes to analyses took place, including those made in response to peer review comments.

11. Results: Line 315-316: Please report the p value in the text for: “...from the Mann-Whitney U test, there was a statistically significant difference between PEI values from the two groups.”

12. Lines 418-430: Please remove the “Author contributions” and “Funding” sections from the manuscript text, and please make sure all information is completely and accurately entered into the relevant sections of the manuscript submission system.

13. Figures and Tables: Please be sure that each text, table, and figure has a descriptive title, and a legend describing the figure/table. We suggest numbering the figures/tables within the supporting information separately from the main text (begin with S1 Table, for example).

14. Table S3: Please report p values in addition to 95% CIs.

15. References: Please check each reference and please use the "Vancouver" style for reference formatting, and see our website for other reference guidelines https://journals.plos.org/plosmedicine/s/submission-guidelines#loc-references

Please check journal title abbreviations. For example, reference 3 should be formatted as PLoS Med.

Comments from Reviewers:

Reviewer #1: I have no additional comments. I congratulate the authors on an excellent paper.

Reviewer #2: The authors have addressed my comments.

Reviewer #3: Comments have been thoroughly addressed

[LINK]

---

## [Editor Report · Decision Letter 3]

9 May 2022

Dear Dr Qian, 

On behalf of my colleagues and the Academic Editor, Mirjam Kretzschmar, I am pleased to inform you that we have agreed to publish your manuscript "Association of pneumococcal carriage in infants with the risk of carriage among their contacts in Nha Trang, Vietnam: A nested cross-sectional survey" (PMEDICINE-D-21-02876R3) in PLOS Medicine.

Please also address the following editorial requests:

-Line 58-59: Please revise this sentence to: “...and that preschool age children may be the largest reservoir for pneumococcal transmission to infants in Nha Trang, Vietnam.”

-Line 68: Please change “vaccinate” to “vaccination”.

-Line 202: Please revise to: "The infant contact study is a cross sectional study and no prospective analysis plan exists. The plan for the current study combining contact and carriage data was conceived during a collaborative meeting that happened after the infant contact and pneumococcal carriage surveys were conducted, but without being informed by the data.” if this is accurate.

-References: Please update reference 4 with the journal publication information (https://doi.org/10.1371/journal.pcbi.100909). Please update the journal title abbreviation for reference 28 to PLoS One.

-Supporting information files list: Please submit files as part of the supporting information in the manuscript submission system. Please do not include links to the external versions of the files (e.g. https://drive.google.com/file…). Please include only one version of the Supplemental Material.

-Figure 2: In the legend, you describe red and green dots to indicate carriage prevalence data, however these points seem to be represented by open circle and “x” symbols, respectively. Please update the legend.

PRESS

Sincerely, 

Caitlin Moyer, Ph.D. 

Associate Editor 

PLOS Medicine